# Peer review of "Study on Water Purification Effect and Operation Parameters of Various Units of Wastewater Circulation"

_water, doi:10.3390/w14111743_

Round 1
Reviewer 1 Report
The work is experimental. Please be clear about the purpose of the research. The layout of the work, i.e. the order of chapters, proportions between particular chapters, the presence of methodological assumptions of work including methods, techniques and research tools, and the interpretation of results is correct and consistent with the accepted rules of editing and performing experimental work. In general, the paper is well written, the results are conclusive and of interest for this topic. The results of the research may be of interest to some readers of Water. It is an important topic and fits in with the current trends and scientific challenges.
Author Response
Review#1:
The work is experimental. Please be clear about the purpose of the research. The layout of the work, i.e. the order of chapters, proportions between particular chapters, the presence of methodological assumptions of work including methods, techniques and research tools, and the interpretation of results is correct and consistent with the accepted rules of editing and performing experimental work. In general, the paper is well written, the results are conclusive and of interest for this topic. The results of the research may be of interest to some readers of Water. It is an important topic and fits in with the current trends and scientific challenges.
Response: Thank you for the reviewer’ suggestions. We have realigned the layout of the work and the methodology, clarifying the purpose of the research in line 14-15 and supplementing the technical and research tools in line 110-112.
Reviewer 2 Report
Manuscript review
MS number: water-1706425
MS title: Study on water purification effect and operation parameters of
various units of wastewater circulation
Evaluation:
(a). Do you recommend that this manuscript be accepted for publication?
No, need major revision and resubmission
(b). The overall length of the text in this manuscript is:
The results and discussion section should be arranged more logically and remove repetitions.
(c). The amount of display material (tables and figures) is:
Both Fig and table numbers should be reduced
(d). Is proper credit given to prior related work?
No, need for more literature review to update the latest information
Author Response
Review#2:Manuscript review, MS number: water-1706425, MS title: Study on water purification effect and operation parameters of various units of wastewater circulation
Evaluation:
(a). Do you recommend that this manuscript be accepted for publication?
No, need major revision and resubmission
Response: Thank you for the reviewer’ suggestions. We have revised as reviewer’s suggested.
(b). The overall length of the text in this manuscript is:
The results and discussion section should be arranged more logically and remove repetitions.
Response: Thank you for the reviewer’ suggestions. We have rearranged the results and discussion sections logically and removed duplicates in line 234-236, 241-242, 265-269.
(c). The amount of display material (tables and figures) is:
Both Fig and table numbers should be reduced
Response: Thank you for the reviewer’ suggestions. We have combined some of the charts (Fig.1-3) in line 179-181 and added the supp. Figure S1 in line 116 to make the potent readers clearly, and no tables in the previous and current versions.
(d). Is proper credit given to prior related work?
No, need for more literature review to update the latest information
Response: Thank you for the reviewer’ suggestions. After checking, our cited references ranged from year 2015 to 2022, only with one reference of year 2009 in line 443. To update the latest information of Chinese’s fish culture status, we have added the latest references in line 38-43.
Reviewer 3 Report
Review Report
Manuscript Title: Study on water purification effect and operation parameters of various units of wastewater circulation
This manuscript describes the preparation in three ways: modified attapulgite (Al@TCAP-N), volcanic stone, and activated carbon for the wastewater circulation system to reduce environmental pollution. This article is well organized; it adequately analyzes the various types of experimental design and sampling procedures, water quality testing, and a better combination of materials for an effective purification strategy; other parameters that affect the wastewater treatments, etc., are extensively studied. Therefore, this present work is recommended for publication in Water Journal. However, the authors can improve this paper considerably to meet the standard of this journal.
Comments:
- Abstract: The abstract is a little bit general; please supplement the meaning and purpose of this research, important characterization, and results in the abstract. Further, please expand the abbreviations such as HRT, TP, and TN in the abstract.
- General comment: the manuscript has typo errors.
- The merits and disadvantages of various water purification strategies concerning wastewater purification in operational cost and sustainability should be elaborated on.
- The ‘conclusion’ should be revised.
Author Response
Review#3:Manuscript Title: Study on water purification effect and operation parameters of various units of wastewater circulation
This manuscript describes the preparation in three ways: modified attapulgite (Al@TCAP-N), volcanic stone, and activated carbon for the wastewater circulation system to reduce environmental pollution. This article is well organized; it adequately analyzes the various types of experimental design and sampling procedures, water quality testing, and a better combination of materials for an effective purification strategy; other parameters that affect the wastewater treatments, etc., are extensively studied. Therefore, this present work is recommended for publication in Water Journal. However, the authors can improve this paper considerably to meet the standard of this journal.
Comments:
1.Abstract: The abstract is a little bit general; please supplement the meaning and purpose of this research, important characterization, and results in the abstract. Further, please expand the abbreviations such as HRT, TP, and TN in the abstract.
Response: Thank you for the reviewer’ suggestions. We supplemented the significance and purpose of the study in line 14-15 as “It is necessary to continuously improve the treatment efficiency of the wastewater treatment devices.”. We also added important representations of method (line 20-22), result and conclusion (line 23-31) as “the removal rate of TP, TN, NH3-N and TSS reached 20%-60%, 20%, 30%-70% and 10%-80%, respectively. The average reduction efficiency of secondary module chlorella and filler on TP, TN, NH3-N, CODMn and TSS was 56.88%, 30.09%, 0.43%, 46.15% and 53.70%, respectively.”, and extended the relevant abbreviations in line 20-23.
2.General comment: the manuscript has typo errors.
Response: Thank you for the reviewer’ suggestions. We have examined the manuscript in detail and corrected the typos in line 122, 184, 207 etc.
3.The merits and disadvantages of various water purification strategies concerning wastewater purification in operational cost and sustainability should be elaborated on.
Response: Thank you for the reviewer’ suggestions. We added a detailed description of the advantages and disadvantages of various wastewater purification strategies in terms of operating costs and sustainability in lines285-292. As “The materials used in the first-level module are inexpensive and could be reused through simple cleaning in this wastewater purification modes, while the secondary module is green and environmentally friendly but occuping a larger pond area and applying for the effective management. It is good to see that the tertiary module can be removed when concerning the operating and management costs, only based on our current limited data. The effective removal rate of constructed wetland (revealed the higher values and to be practical in filed culture) need to be further studied [19].”.
4.The ‘conclusion’ should be revised.
Response: Thank you for the reviewer’ suggestions. We revised the "Conclusions" as “The second unit obtained the enhanced removal effect when plus adding microorgan-isms and activated sludge, while the third unit could be adjusted according to the cost and labor management.” in line 332-345.
Round 2
Reviewer 2 Report
N/A
Reviewer 3 Report
The revised manuscript can be accepted for publication